# Cuticular Chemistry of the Queensland Fruit Fly *Bactrocera tryoni* (Froggatt)

**DOI:** 10.3390/molecules25184185

**Published:** 2020-09-12

**Authors:** Soo J. Park, Gunjan Pandey, Cynthia Castro-Vargas, John G. Oakeshott, Phillip W. Taylor, Vivian Mendez

**Affiliations:** 1Applied BioSciences, Macquarie University, North Ryde, NSW 2109, Australia; Gunjan.Pandey@csiro.au (G.P.); Cynthia.Castro-Vargas@csiro.au (C.C.-V.); John.Oakeshott@csiro.au (J.G.O.); phil.taylor@mq.edu.au (P.W.T.); vivian.mendez@mq.edu.au (V.M.); 2Australian Research Council Centre for Fruit Fly Biosecurity Innovation, Macquarie University, North Ryde, NSW 2109, Australia; 3Commonwealth Scientific and Industrial Research Organisation Land and Water, Black Mountain, Acton, ACT 2601, Australia

**Keywords:** cuticular hydrocarbons, cuticle, chemical communication, GC-MS, methyl branched alkanes, chemical ecology, volatiles

## Abstract

The cuticular layer of the insect exoskeleton contains diverse compounds that serve important biological functions, including the maintenance of homeostasis by protecting against water loss, protection from injury, pathogens and insecticides, and communication. *Bactrocera tryoni* (Froggatt) is the most destructive pest of fruit production in Australia, yet there are no published accounts of this species’ cuticular chemistry. We here provide a comprehensive description of *B. tryoni* cuticular chemistry. We used gas chromatography-mass spectrometry to identify and characterize compounds in hexane extracts of *B. tryoni* adults reared from larvae in naturally infested fruits. The compounds found included spiroacetals, aliphatic amides, saturated/unsaturated and methyl branched C_12_ to C_20_ chain esters and C_29_ to C_33_ normal and methyl-branched alkanes. The spiroacetals and esters were found to be specific to mature females, while the amides were found in both sexes. Normal and methyl-branched alkanes were qualitatively the same in all age and sex groups but some of the alkanes differed in amounts (as estimated from internal standard-normalized peak areas) between mature males and females, as well as between mature and immature flies. This study provides essential foundations for studies investigating the functions of cuticular chemistry in this economically important species.

## 1. Introduction

The cuticular layer of the insect exoskeleton contains a range of mostly aliphatic compounds, including normal and branched alkanes, alkenes, saturated and unsaturated esters, alcohols, saturated and unsaturated fatty acids, ketones, and aldehydes [1]. Cuticular hydrocarbons usually contain 20 to 50 carbons, and compounds with other functional groups vary from 12 to 54 carbons [2,3,4]. A primary function of cuticular hydrocarbons is to protect against desiccation [5,6], injury, and infection [7,8,9,10,11,12]. Cuticular compounds, including hydrocarbons, are also commonly important for chemical communication [2,5,6,13], including species recognition [14,15,16,17], mimicry [18], and as pheromones [19] in diverse insect taxa. For example, some cuticular hydrocarbons serve as sex pheromones in house fly [20], the circumboreal fly [21], moths [22], bees [23] and the cowpea weevil [24]. Cuticular hydrocarbons also serve as aggregation pheromones in some insects, including *Drosophila* [25], termites [26] and cockroaches [27]. Sexual selection has been a driving force for the evolution of sexual dimorphism in animals [28], and many insect taxa exhibit sexual dimorphism in cuticular chemistry [29]. For example, sexually dimorphic cuticular hydrocarbons have been found in *Drosophila* [30,31,32] and have been implicated in female attractiveness and male mating success [21].

Tephritid fruit flies are amongst the world’s most economically damaging insect pests [33]. Some aspects of tephritid fruit fly semiochemistry have received significant attention, particularly the pheromones [34,35] they use to attract mates and for aggregation and the particular compounds found in fruit, food and certain flowers to which they are attracted [36,37,38,39,40,41,42,43,44,45,46,47]. Some work has also been performed on fruit fly cuticular chemistry because their cuticular chemical profiles tend to be highly species-specific [48,49] and have been used to resolve species, cryptic species and geographic variation in larvae [50,51,52] and adults [53,54,55,56,57,58]. Beyond their use as chemotaxonomic tools, however, relatively little work has been performed on tephritid cuticular compounds. In an important recent exception, allyl-2,6-dimethoxyphenol has been proposed as a short-range male attractant in *Bactrocera dorsalis* [59]. Most cuticular compounds are aliphatic, so this case is also notable for its involvement of an aromatic compound.

In Australia and in some Pacific Islands, the Queensland fruit fly, *Bactrocera tryoni* (Froggatt), is an economically important pest of horticultural crops [60,61,62]. This species causes significant economic loss by damaging crops [63] and by limiting market access [64]. While rectal gland and volatile emission chemistry of *B. tryoni* has been documented [65,66,67,68,69,70], the cuticular chemistry of this species has not. Given what is known for other insects, the composition of its cuticular chemical profile is likely to be relevant to homeostasis, protection from pathogens, injury and insecticides [71], and chemical communication. Understanding elements of cuticular chemistry related to homeostasis may help to understand abiotic factors mediating bioclimatic potential of *B. tryoni* [72] and effects of domestication, sex and age on desiccation resistance [73,74], and may also be important for understanding environmental and sexual competence of sterile *B. tryoni* released in sterile insect technique (SIT) programs to control pest populations [75,76,77]. To address this knowledge gap, and to provide foundations for subsequent functional studies, the present study reports a qualitative description of *B. tryoni* cuticular chemistry and identifies qualitative and quantitative variation (the latter estimated from internal standard-normalized peak areas) related to maturity and sex. *Bactrocera tryoni* specimens were obtained as larvae in infested fruits and cuticle extracts of emerged adults were analyzed by gas chromatography-mass spectrometry (GC-MS).

## 2. Results

### Cuticular Chemistry and Statistical Analysis

The identified compounds are all aliphatic; no trace of aromatic compounds was found. Typical chromatograms of both immature and mature female and male *B. tryoni* are shown in Figure 1. The chromatogram sections of shorter (**A**) and longer retention time compounds (**B**) of *B. tryoni* adults are shown in Figure 2, where a typical chromatogram of a female is presented because the chromatogram includes all the compounds that are also found in immature and mature females and males. A range of non-alkanes, including spiroacetals, amides and esters and an assortment of C_29_ to C_33_ methyl-branched alkanes represent the cuticular chemistry of wild *B. tryoni* adults.

The identities of **22** of the **32** non-alkane compounds found were confirmed with authentic standards and the other ten were tentatively identified by comparison to fragment patterns in NIST libraries (Table 1). Two of the compounds were 6,6-membered ring spiroacetals (compounds **A2** and **A5** in Table 1 and Figure 2), four were aliphatic amides (compounds **A1**, **A3**, **A4** and **A6** in Table 1 and Figure 2), and the remaining **26** were all esters of saturated/unsaturated and methyl branched saturated fatty acids (compounds **A7**–**A32** in Table 1 and Figure 2). The spiroacetals and saturated/unsaturated and branched saturated esters were found to be specific to mature females, while the amides were found in mature flies of both sexes (Table 1). In mature females, ethyl esters of saturated or unsaturated C_12_, C_14_, C_16_ and C_18_ are the most abundant, while methyl and propyl esters, and branched saturated fatty acid esters are minor and trace, respectively. Methyl positions in the branched fatty acid esters are ambiguous, because trace amounts of the compounds made it difficult for further analyses. The amount of *N*-(3-methylbutyl)isobutyramide (**A6**) was about 2.5 times larger in mature male than in mature females (*p* < 0.05, *t*-test), but the amount of *N*-(3-methylbutyl)propanamide (**A4**) was about 3.2 times larger in mature females than in mature males (*p* < 0.05, *t*-test). The differences in the amounts of the other amides, *N*-(3-methylbutyl)acetamide (**A1**) and *N*-(2-methylbutyl)propanamide (**A3**) were not significant (*p* > 0.05 for all, *t*-test). The total amount of the amides was 2.6 times larger in mature females than in mature males (*p* < 0.05, *t*-test). The results are illustrated in Figure 3.

The 34 tentatively identified hydrocarbons, all methyl-branched alkanes with C_29_ to C_33_ carbon backbones, are summarized in Table 2. Unsaturated hydrocarbons were not detected. Most are mono- or dimethylalkanes, with only a few trimethylalkanes found. Monomethyl branches appeared exclusively at odd carbon positions in odd carbon alkanes and at even carbon positions in even carbon alkanes. Comparisons in the amounts of individual hydrocarbons between sexes are illustrated in Figure 4. The most abundant alkanes were mono- and dimethylhentriacontane (C_31_) isomers (compounds **B18** to **B24** in Figure 2B and Figure 4), with the 11-, 13- and 15-methylhentriacontanes (**B18**) appearing at the highest intensity in chromatograms. Although dimethyl branches were separated by 1, 3, 5, 7, 9, 11, or 13 methylene groups, a majority of dimethyl branches were separated by 3 and 5 methylenes. Trimethyl branches were separated by [3,3], [3,5] or [3,11] methylenes. 

There were significant effects of sexual maturity, sex and the interaction between these two variables on the amounts of the normal and methyl branched alkanes (sexual maturity *F*_1,2276_ = 81.90, *p* < 0.0001; *F*_1,2276_ = 10.56, *p* < 0.005; sexual maturity * sex *F*_1,2276_ = 4.10, *p* < 0.05, respectively). The amounts of alkanes in mature flies were 2.3 (±1.3) times larger than those in immature flies on average. The amounts of compounds **B1**, **B3**, **B4**, **B5**, **B6**, **B7**, **B9**, **B10**, **B11**, **B12**, **B13**, **B18**, **B19**, **B20**, **B21**, **B22**, **B22**, and **B24** were higher in mature females than in mature males (*p* < 0.05, *t*-test). The amounts of these compounds were 1.7 (±0.5) times greater in mature females than in mature males on average. There was no significant difference in the amount of hydrocarbons between immature females and immature males (*p* > 0.05, *t*-test).

## 3. Discussion

The present study finds that the *n*-hexane-extracted cuticular chemistry of *B. tryoni* includes a complex mixture of at least 66 compounds, including two spiroacetals, four aliphatic amides, 26 saturated/unsaturated C_12_ to C_20_ methyl, ethyl and propyl esters and 34 methyl branched saturated alkanes with a range of C_29_ to C_33_ carbon backbones. A previous study reported 14 cuticular compounds in *B. tryoni*, including five fatty acid esters, two siloxanes and seven methyl branched alkanes [98]. We did not detect the siloxanes or the methyl branched alkanes, which are all shorter than the alkanes found in the present study. We suspect that occurrence of the siloxanes in the previous work may reflect impurities, and incorrect assignments may have been given for the shorter methyl branched alkanes. The differences might have been also caused by technical differences; for example the previous work extracted cuticular compounds in methanol for 20 min [98], while the present study extracted in *n*-hexane for 3 min. The solvent choice of *n*-hexane was based on anticipated polarities of insect cuticular compounds that are generally less or non-polar [99]. The method used in the present study is more similar to methods widely used in studies of the cuticular hydrocarbons of other tephritids [53,54,55,56,57,58,59,100].

We found sexual dimorphism in several aspects of the cuticular chemistry of *B. tryoni*. In particular, our data suggest that spiroacetals and esters are specific to mature females. Sex-related differences in cuticular chemistry have also been reported in *B. dorsalis*, in which mature males have 7-monoenes that are absent from mature females and immatures of both sexes [100]. We also found quantitative differences between the sexes in their cuticular amides, which overall were more abundant in mature females than mature males, even though they are known to be particularly abundant in the rectal glands of mature males. Although the amides from male rectal glands have been suggested to function as male sex pheromones [65,66,68], the functions of these compounds on the cuticle, and in rectal glands of mature females, are not known. Our results indicate that more work is now needed on their functions in both sexes, both in rectal glands and on the cuticle.

Most of the non-alkane components which we find in cuticular extracts of *B. tryoni* have also been reported previously in rectal gland extracts of this species, with some of the same sex differences also evident [65,67,68,69]. In particular, the spiroacetals, amides and saturated/unsaturated esters we found in female cuticles had also been reported in female rectal glands. The presence of minute amounts of the amides and absence of the spiroacetals found on male cuticles also matches the earlier findings for male rectal glands, although the previous work also reported shorter chain esters (C_5_ or less) in male rectal glands that were not found on cuticles. Notwithstanding the difference in the esters, the similarities between the two extract types otherwise suggest that at least some of the non-alkane components on the cuticle could originate in the rectal glands and be distributed over the body when grooming, as has been described in some social insects [101,102]. The differences in the esters and the quantitative differences in the amides noted above might in part reflect differences in volatility of the various compounds.

The C_12_ to C_20_ esters we found in cuticle of mature female *B. tryoni* have also been reported in the cuticle of females in five *B. dorsalis* complex species, but at different relative abundances between the species, suggesting that they may have a role in species recognition [58,59]. On the other hand, the methyl branched fatty acid esters found in the present study have not been previously reported in either the emission profile or rectal gland extracts of *B. tryoni* [67], or other fruit flies. Methyl branched fatty acid methyl, propyl or longer alkyl esters occur in other arthropods, such as spiders [103,104], but, to the best of our knowledge, specific roles have not yet been identified for them in any species. Branched fatty acid esters are biosynthetically feasible [105,106] and fatty acids are also the precursors of other cuticular compounds and hydrocarbons [2]. Hence, the branched esters found in this study may be intermediate products in the biosynthesis of other cuticular compounds.

While most of the C_29_ to C_33_ methyl branched alkanes detected in the present study have also been reported in other organisms (see references in Table 2), several of them have not previously been described in tephritids. Branched alkanes and/or unsaturated hydrocarbons have lower melting points than the corresponding normal alkanes, allowing the waxy layer of insect cuticles to be flexible over a wide range of ambient temperatures [107,108]. Long chain hydrocarbons, and in particular branched and unsaturated alkanes, have also been linked to desiccation resistance in a variety of insects [109,110,111]. However, to what extent that applies in *B. tryoni* adults remains unclear. While qualitatively similar across the sexes and age groups studied, we found higher amounts of alkanes (as estimated from internal-standard-normalised peak areas) in mature than immature flies but, at least in domesticated flies from colonies maintained on artificial diets, desiccation resistance of *B. tryoni* decreases with age [73].

Several normal and methyl branched C_28_ to C_40_ alkanes have also been reported previously in two closely related taxa in the *Bactrocera dorsalis* complex, but as different isomers and in different amounts between the two [57]. Such variation in cuticular hydrocarbon chemistry has also been reported among species of another tephritid genus, *Anastrepha* [53,55,56,58,100], and even among populations within *Ceratitis rosa* [55]. As with the esters above, these taxon-specific cuticular hydrocarbon signatures may play a role in taxon recognition in nature, but they may also make excellent tools for taxonomists [53,55,56,57,58,100]. Their use in taxonomy is directly relevant to the *B. tryoni* complex which, despite its importance as a pest, is not well understood taxonomically. The *B. tryoni* complex includes three other taxa; *B. neohumeralis*, *B. aquilonis* and *B. melas* [60]. *Bactrocera tryoni* is clearly differentiated from *B. neohumeralis* in the timing of mating behavior [112], but the species status of *B. aquilonis* and *B. melas* is still debated [113] and the four taxa differ in their pest status and quarantine restrictions in various jurisdictions [113]. However, before cuticular chemistry can be used to resolve such taxonomic issues it will be important to investigate the extent to which it can be influenced by diet, maturity and the physical environment [114,115,116]. Given the population differences found in *Ceratitis rosa* [55], it will also be important to determine whether cuticular chemistry varies between geographic regions and during the course of domestication of species in the *B. tryoni* complex.

In summary, the present study provides the first detailed description of the cuticular chemistry of *B. tryoni*, and finds some clear qualitative and quantitative differences as flies mature and between the sexes. Our findings provide a foundation for studies addressing the roles of cuticular chemistry in functions such as desiccation resistance, protection from pathogens and injury, and chemical communication, as well as its potential application in resolving the taxonomy of the *B. tryoni* complex.

## 4. Materials and Methods

### 4.1. Chemicals

Ethyl dodecanoate (ethyl laurate), ethyl tridecanoate, propyl dodecanoate, methyl tetradecanoate, ethyl (*E)*-9-tetradecenoate (ethyl myristelaidate), ethyl (*Z*)-9-tetradecenoate (ethyl myristoleate), ethyl tetradecanoate (ethyl myristate), ethyl pentadecanoate, methyl (*Z*)-9-hexadecenoate, methyl hexadecanoate, ethyl (*E*)-9-hexadecenoate (ethyl palmitolelaidate) (*Z*)-9-hexadecenoate (ethyl palmitoleate), ethyl hexadecanoate (ethyl palmitate), ethyl (*Z,Z*)-octadeca-9,12-dienoate (ethyl linoleate), ethyl (*Z*)-9-octadecenoate (ethyl oleate), ethyl (*E*)-9-octadecenoate (ethyl elaidate), ethyl octadecanoate, hexadecane and straight-chain C_8_–C_40_ alkane standards were purchased from Sigma-Aldrich (St. Louis, MO, USA), Nu-Chek-Prep and INC (Minneapolis, MN, USA) or Alfa-Aesar. *N*-(3-methylbutyl)acetamide, *N*-(2-methylbutyl)propanamide, *N*-(3-methylbutyl)propanamide, *N*-(3-methylbutyl)isobutyramide were synthesized by reactions of an appropriate amine and an acid anhydride in water. Synthetic details are presented in Appendix A. 2,8-Dimethyl-1,7-dioxaspiro[5.5]undecane was kindly provided by Ms. Sally Noushini.

### 4.2. Origin of Flies

*Bactrocera tryoni* larvae were collected from infested loquat fruits from a tree located in Marsfield NSW Australia (33.766080 S, 151.100722 E). A 500 mL plastic container with approximately 300 g of infested loquat fruits was placed on approximately a 1 cm deep layer of vermiculite in a 12.5 L plastic box for larvae to complete development and exit the fruit to pupate. Pupae were sieved and moved to a fine mesh cage (47.5 × 47.5 × 47.5 cm) (Megaview Bugdorm 4S4545, Taiwan) where the flies emerged. Identity of *B. tryoni* was confirmed by examining emerged flies under a stereomicroscope using the key to tropical tephritid fruit flies [117] and Jane Royer kindly provided additional confirmation for the identity of *B. tryoni*. Adults were separated by sex within three days of emergence, when still sexually immature [118], and thereafter kept as single-sex cohorts of 80 flies in mesh cages (30.5 × 30.5 × 30.5 cm) (Megaview Bugdorm 4S4545, Taiwan). Adult flies were provided with food by coating a small area of the top of mesh cage with a paste prepared by mixing 15 g sugar, 5 g hydrolysate yeast (MP Biomedicals LLC) and 4 mL tap water. Water was provided by placing an inverted vial (6 cm height, 4 cm diameter) full of water on a sponge on the top of the mesh cage. All cages were maintained in a controlled environment room at 25 ± 0.5 °C, 65 ± 5% RH and photoperiod of 11.5:0.5:11.5:0.5 light: dusk: dark: dawn.

### 4.3. Extraction of Cuticular Compounds

Fifteen two-day old sexually immature flies of each sex and fifteen 20-day old sexually mature virgin flies of each sex were used for the extraction of cuticular compounds. The flies were killed by placing them in a 5 mL plastic vial on dry ice. Frozen flies were allowed to defrost at room temperature for three minutes immediately before the following extraction procedure. *n*-Hexane was chosen as the extraction solvent because many cuticular compounds found in other tephritids are less, or non-polar [53,54,55,56,57,58,59,100]. A single fly was immersed in 400 µL of *n*-hexane that contained 1.5 μg/mL *n*-hexadecane (Sigma-Aldrich, St. Louis, MO, USA) in a 1.5 mL clear glass vial. *n*-Hexadecane was used as an internal standard to normalize peak areas for comparisons between groups. The vial containing a fly was allowed to stand for three minutes at room temperature, then the fly was removed from the *n*-hexane extract. If *n*-hexane extracts contained aqueous droplets, the droplets were removed by adding sodium sulfate (Na_2_SO_4_) (Sigma-Aldrich, St. Louis, MO, USA) and gravity filtration. If samples contained solid organic matter, the solid was removed by gravity filtration. Gravity filtration was achieved by filtering the sample through a glass wool plug on the neck of a Pasteur pipette. *n*-Hexane extracts were concentrated to 120 µL under a gentle stream of nitrogen gas and transferred to a glass insert (150 µL) in a clear 1.5 mL GC vial. The samples were stored at 4 °C until analyzed.

### 4.4. Gas Chromatography Mass Spectrometry (GC-MS) Analysis

GC-MS analysis was carried out on a Shimadzu GCMS TQ8040 spectrometer equipped with a split/splitless injector and SH Rtx-5MS (30 m × 0.25 mm, 0.25 µm film) fused silica capillary column. The carrier gas was helium (99.999%) (BOC, North Ryde, NSW, Australia) at a flow rate of 1.0 mL/min. An aliquot of 1 µL sample was injected at splitless mode where the injector temperature was 270 °C. The temperature program was set initially at 50 °C (1 min), increased to 280 °C at a rate of 10 °C/min, then increased to 302 °C at a rate of 2 °C/min. The ion source and transfer line temperatures were 200 and 290 °C, respectively. The ionization method was electron impact at a voltage of 70 eV. The spectra were obtained over a mass range of *m*/*z* 47–650. The data were analyzed by Shimadzu GCMS Postrun program (Shimadzu, Kyoto, Japan) and compared with the mass fragmentation patterns in NIST libraries (NIST17-1, NIST17-2), and authentic standards. Retention indices (KI) were calculated with the Kovats retention index equation [119] and compared with KI values published in the literature. The structures of methyl-branched alkanes were assigned by using methods described in previous studies [120,121,122]. Briefly, the chain length and number of methyl groups of a methyl-branched alkane were established by examining an equivalent chain length and molar mass. The position of a methyl group was then assigned by examining fragment ions.

### 4.5. Comparison between Groups

The data were normalized by dividing the GC peak areas of individual components by the peak areas of the internal standard. The data were not normally distributed and hence were log-transformed for statistical analyses. However, the raw data were used to generate the graphs. The data for normal and branched alkanes were analyzed by ANOVA to test for effects of sex and maturity on the amounts of compounds. The individual amides and alkanes were compared between sexes by *t*-test.

## Figures and Tables

**Figure 1 molecules-25-04185-f001:**
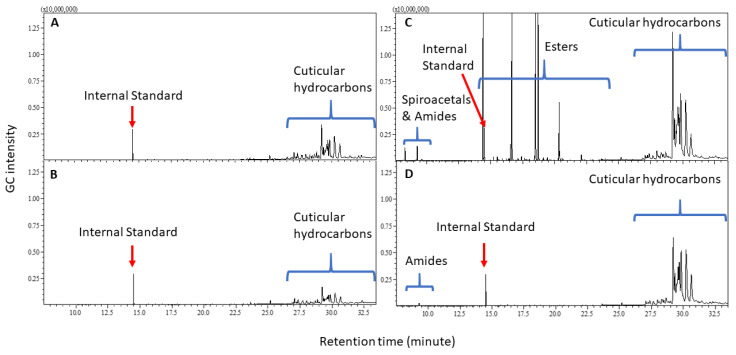
Typical chromatograms of hexane extracts of immature and mature female and male *B. tryoni*. (**A**) immature female; (**B**) immature male; (**C**) mature female; (**D**) mature male.

**Figure 2 molecules-25-04185-f002:**
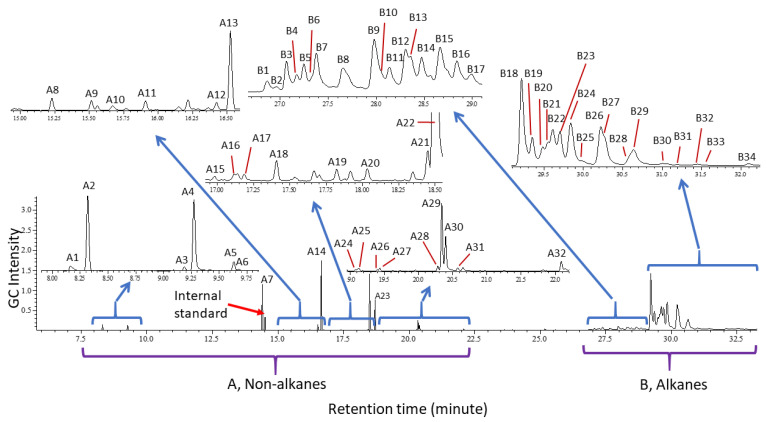
Representative chromatogram sections. (**A**) Non-alkanes in a chromatogram of a mature female *B. tryoni*, which includes all compounds found in mature and immature males and females; (**B**) Hydrocarbons (alkanes) section of chromatogram from a mature female *B. tryoni*. Note that compounds in B are qualitatively identical in immature and mature females and males.

**Figure 3 molecules-25-04185-f003:**
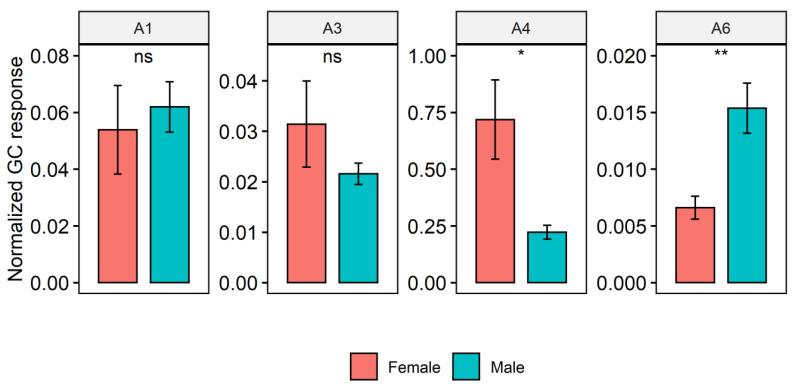
Internal standard-normalized peak areas of *N*-(3-methylbutyl)acetamide (**A1**), *N*-(2-methylbutyl)propanamide (**A3**), *N*-(3-methylbutyl)propenamide (**A4**) and *N*-(3-methylbutyl)isobutyramide (**A6**) in section A in Figure 2 in female and male *B. tryoni*. Standardized peak areas were obtained by dividing the peak area of a compound by the peak area of the *n*-hexadecane internal standard. Error bars represent standard errors. The results of *t*-test comparisons between the sexes are shown (ns is not significant; * *p* < 0.05, ** *p* < 0.01).

**Figure 4 molecules-25-04185-f004:**
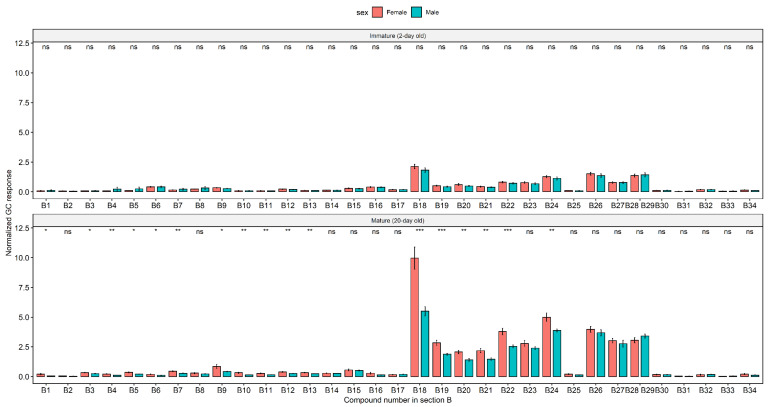
Internal standard-normalized peak areas of individual cuticular hydrocarbons (the alkanes in section B in Figure 2) in the two sexes of *B. tryoni*. Standardized peak areas were obtained by dividing the peak area of a compound by the peak area of the *n*-hexadecane internal standard. **B28** and **B29** co-eluted and hence the sum of their amounts are presented together. Error bars represent standard errors. The results of *t*-test comparisons between the sexes are shown (ns is not significant; * *p* < 0.05; ***p* < 0.01; *** *p* < 0.001).

**Table 1 molecules-25-04185-t001:** The compounds identified in hexane washes of *B. tryoni* that eluted early in chromatograms (section A in Figure 2).

No	Identity	MM	KI	Ref.KI (Ref)	Characteristic/Diagnostic EI Ions
A1 *	*N*-(3-Methylbutyl)acetamide	129.12	1131	1137 [78]	129 (M^+^), 114, 86, 73, 60 (CH_3_COHNH_2_^+^)
A2	2,8-Dimethyl-1,7-dioxaspiro[5,5]undecane	184.15	1140	1147 [78]	184 (M^+^), 140, 115/112 (M-C_5_H_8_/C_5_H_8_O), 97, 69
A3 *	*N*-(2-Methylbutyl)propanamide	143.13	1198		143 (M^+^), 114, 86, 74, 57
A4 *	*N*-(3-Methylbutyl)propanamide	143.13	1204		143 (M^+^), 128 (M-CH_3_), 114 (M-C_2_H_5_), 100, 87, 74, 57
A5 #	2-Ethyl-8-methyl-1,7-dioxaspiro[5,5]undecane	198.16	1230	1237 [78]	198 (M^+^), 169, 129/126 (M-C_5_H_8_/C_5_H_8_O), 115/112 (C_6_H_10_/C_6_H_10_O), 97, 83, 69, 55
A6 *	*N*-(3-Methylbutyl)isobutyramide	157.15	1233		157 (M^+^), 142, 101, 71, 57
A7	Ethyl dodecanoate (ethyl laurate)	228.38	1591	1593 [79]	228 (M^+^), 183, 157, 115, 101, 88, 73, 70, 60 (CH_3_CO=OH^+^)
A8 #	Ethyl 6-methyldodecanoate	242.22	1662		242 (M^+^), 213, 199, 185, 157, 143, 101, 88, 83, 70, 55
A9	Propyl dodecanoate	242.22	1680	1685 [80]	242 (M^+^), 201, 183, 157, 143, 115, 102, 61 (C_3_H_7_OH_2_^+^, base peak)
A10	Ethyl tridecanoate	242.22	1691	1695 [81]	242 (M^+^), 199, 197, 157, 101, 88
A11	Methyl tetradecanoate	242.22	1722	1724 [82]	242 (M^+^), 157, 143, 101, 87, 74
A12	Ethyl (*E*)-9-tetradecenoate (ethyl myristolaidate)	254.22	1769		254 (M^+^), 208/209 (loss of EtOH/EtO), 166, 124, 88, 55
A13	Ethyl (*Z*)-9-tetradecenoate (ethyl myristoleate)	254.22	1778		254 (M^+^), 208/209 (loss of EtOH/EtO), 166, 124, 88, 55
A14	Ethyl tetradecanoate (ethyl myristate)	256.43	1790	1793 [80]	256 (M^+^), 213, 157, 101, 88
A15 #	Ethyl 4-methyltetradecanoate	270.26	1836		270 (M^+^), 213, 101 (M-C_12_H_25_, base peak), 88
A16 #	Ethyl 12-methyltetradecanoate	270.26	1862		270 (M^+^), 227, 213, 157, 101, 88
A17 #	Propyl tetradecanoate	270.26	1887	1893 [83]	270 (M^+^), 229, 211, 172, 129, 102, 61 (C_3_H_7_OH_2_^+^, base peak)
A18	Ethyl pentadecanoate	270.26	1890	1897 [84]	270 (M^+^), 227, 199, 157, 101, 88
A19	Methyl (*Z*)-9-hexadecenoate	268.44	1902	1909 [78]	268 (M^+^), 236/237 (loss of MeOH/MeO), 194, 152, 96, 74, 55
A20	Methyl hexadecanoate	270.26	1923	1927 [85]	270 (M^+^), 227, 143, 87, 74
A21	Ethyl (*E*)-9-hexadecenoate (ethyl palmitelaidate)	282.26	1965		282 (M^+^), 236/237 (loss of EtOH/EtO), 194, 152, 96, 88, 69, 55
A22	Ethyl (*Z*)-9-hexadecenoate (ethyl plamitoleate)	282.26	1970	1975 [86]	282 (M^+^), 236/237 (loss of EtOH/EtO), 194, 152, 96, 88, 69, 55
A23	Ethyl hexadecanoate (ethyl palmitate)	284.27	1990	1993 [80]	284 (M^+^), 241, 157, 101, 88
A24 #	Ethyl 15-methylhexadecanoate	298.29	2029		298 (M^+^), 255, 157 (M-C_10_H_21_), 101, 88
A25 #	Ethyl 4-methylhexadecanoate	298.29	2035		298 (M^+^), 241 (M-C_4_H_9_), 101 (base peak), 88
A26 #	Ethyl 14-methylhexadecanoate	298.29	2062		298 (M^+^), 269, 255, 241, 199, 157, 101, 88
A27 #	Propyl 9-hexadecenoate	296.27	2067		296 (M^+^) 281, 237, 194 (M-C_3_H_7_COOCH_2_)
A28	Ethyl (*Z,Z*)-octadeca-9,12-dienoate (ethyl linoleate)	308.27	2158	2155 [87]	308 (M^+^), 262/263 (loss of EtOH/EtO), 178, 135, 95, 81
A29	Ethyl (*Z*)-9-octadecenoate (ethyl oleate)	310.29	2168	2168 [88]	310 (M^+^), 264/265 (loss of EtOH/EtO), 222, 180, 97, 55
A30	Ethyl (*E*)-9-octadecenoate (ethyl elaidate)	310.29	2171	2174 [89]	310 (M^+^), 264/265 (loss of EtOH/EtO), 222, 180, 97, 55
A31	Ethyl octadecanoate	312.54	2190	2191 [90]	312 (M^+^), 269, 157, 101, 88
A32 #	Ethyl 11-eicosenoate	338.57	2366		338 (M^+^), 292/293 (M-EtOH/EtO), 250, 208, 97, 55

MM: molecular mass, KI: Kovats’ retention index, Ref.KI (Ref): reference KI if available for a similar column type, active phase and temperature conditions, with references in parentheses; * indicates that the compound is present in both sexes of mature adults, all other compounds being mature female specific; # indicates the compound was only tentatively identified.

**Table 2 molecules-25-04185-t002:** The tentatively identified cuticular hydrocarbons found in *n*-hexane washes of *B. tryoni* (section B in Figure 1).

No	Identity	MM	KI	Ref.KI (Ref)	Characteristic/Diagnostic EI Ions
B1	11-; 13-; 15-MeC_29_	422.82	2929	2932 [91]	280/281,168/169; 252/253, 196/197; 224/225(s)
B2	7-MeC_29_	422.82	2946	2940 [51]	336/337, 112/113
B3	5-MeC_29_	422.82	2952	2949 [91]	364/365, 84/85
B4	9,13-DiMeC_29_	436.50	2966	2963 [91]	322/323, 140/141, 252/253, 210/211
B5	7,11-DiMeC_29_	436.50	2970		350/351, 112/113, 280/281, 182/183
B6	3-MeC_29_	422.82	2976	2973 [91]	392/393, 56/57
B7	5,11-DiMeC_29_; 5,13-DiMeC_29_	436.50	2986	2983 [91]	378/379, 84/85,280/281, 182/183; 378/379, 84/85, 280/281, 210/211
B8	4, *x*, 22-TriMeC_29_ (*x* = 14 or 16)	450.52	3009		392/393, 84/85, 252/253, 224/225, 336/337, 126/127
B9	12-Me; 14-MeC_30_	436.50	3025	3031 [92]	280/281,182/183; 252/253, 210/211
B10	8-MeC_30_	436.50	3034	3040 [93]	336/337, 126/127
B11	6-MeC_30_	436.50	3041	3045 [93]	364/365, 98/99
B12	4-MeC_30_	436.50	3055	3065 [93]	392/393, 70/71
B13	8,12-DiMeC_30_; 8,14-DiMeC_30_	450.52	3061	3064 [94]	350/351, 126/127, 280/281, 196/197; 350/351, 126/127, 253/252, 225.224
B14	6,14-DiMeC_30_; 6,12-DiMeC_30_	450.50	3071		378/379, 98/99, 252/253, 224/225; 378/379, 98/99, 280/281, 196/197
B15	4,12-DiMeC_30_; 4,14-DiMeC_30_; 4,20-DiMeC_30_	450.52	3088	3098 [94]	406/407, 70/71, 280/281, 196/197; 406/407, 70/71, 225/224, 253/252; 406/407, 70/71, 309/308, 169/168
B16	*n*-C_31_	436.50	3100		436
B17	4,8,12-TriMeC_30_; 4,8,14-TriMeC_30_; 4,8,20-TriMeC_30_	464.53	3115		70/71, 420/421, 350/351, 140/141, 280/291, 210/211; 70/71, 420/421, 350/351, 140/141, 252/253, 238/239; 70/71, 420/421, 350/351, 140/141, 322/323, 168/169
B18	11-; 13-; 15-MeC_31_	450.52	3129	3130 [92]	308/309, 168/169; 280/281, 196/197; 250/251, 224/225
B19	7-MeC_31_; 9-MeC_31_	450.52	3137	3140 [93]	364/365, 112/113; 336/337, 141/140
B20	5-MeC_31_	450.52	3147	3150 [93]	392/393, 84/85
B21	11,15-DiMeC_31_	464.53	3153	3155 [93]	322/323, 168/169, 252/253, 238/239
B22	9,13-DiMeC_31_; 9,15-DiMeC_31_; 11,13-DiMeC_31_; 13,15-DiMeC_31_	464.53	3157	3159 [92]	350/351, 140/141, 280/281, 210/211; 350/351, 140/141, 252/253, 238/239; 322/323, 168/169, 280/281, 210/211; 294/295, 196/197, 252/253, 238/239
B23	7,13-DiMeC_31_;7,15-DiMeC_31_	464.53	3164		378/379, 112/113, 280/281, 210/211; 378/379, 112/113, 252/253, 238/239
B24	3-MeC_31_	450.52	3170	3172 [91]	420/421, 56/57
B25	5,11-DiMeC_31_; 5,13-DiMeC_31_	464.53	3178	3180 [95]	406/407, 84/85, 182/183, 308/309; 406/407, 84/85, 280/281, 210/211
B26	12-; 14-; 16-MeC_32_	464.53	3223	3225 [57]	308/309, 182/183; 280/281, 210/211; 252/253, 238/239
B27	8-; 10-MeC_32_	464.53	3225	3225 [57]	364/365, 126/127; 336/337, 154/155
B28	10,14-DiMeC_32_	478.55	3233	3254 [94]	350/351, 154/155, 280/281, 224/225
B29	8,12-DiMeC_32_; 8,14-DiMeC_32_; 8,16-DiMeC_32_	478.55	3257	3263 [96]	378/379, 126/127, 308/309, 196/197; 378/379, 126/127, 280/281, 224/225; 378/379, 126/127, 252/253 (s)
B30	4-MeC_32_	464.53	3267	3265 [97]	420/421, 70/71
B31	6,16-DiMeC_32_	478.55	3285		406/407, 98/99, 252/253
B32	9-; 11-MeC_33_	478.55	3327	3335 [93]	364/365, 140/141; 336/337, 168/169
B33	13-; 15-; 17-MeC_33_	478.55	3330	3335 [93]	308/309, 196/197; 280/281, 224/225; 252/253 (s)
B34	9,23-DiMeC_33_	492.56	3359		378/379, 140/141, 350/351, 168/169

MM: molecular mass; KI: Kovats retention index; Ref.KI (Ref): reference KI if available with references in parenthesis; (s) indicates ions from symmetrical structures.

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
