# Peer review of "Cuticular Chemistry of the Queensland Fruit Fly Bactrocera tryoni (Froggatt)"

_molecules, 2020, doi:10.3390/molecules25184185_

Round 1

Reviewer 1 Report

The paper is concerning the chemical analysis of the cuticular hexane extract of Bactrocera tryoni because the constituents of the exoskeleton have a role on the biological functions such as homeostasis, communication etc... GC was used to identify the constituents, at least tentatively. Results are limited because they are mainly represented by the list of the identified constituents with a general discussion. Some additional experiments exploring the biological implications of such differences could be carried out. The manuscript in the present form  is not sufficiently interesting to the readers and originality and scientific soundness are poor. 

Reviewer 2 Report

In the introduction, I put some suggestions in the document in PDF.

In this work it is observed, due to the characteristics of the method, that samples were analyzed at the trace level, difficult to detect, even with this equipment, I suppose they are quantities between ng-pg.

The chromatograms show peaks with low resolution, it is recommended to work more on the separation of the peaks. Change to a column of smaller diameter, or greater length. What I suggest is that the compounds that have good resolution be taken and those are the ones that are reported. Although, I understand that the identification was done properly, with the comparison of their mass spectra, retention index, and 23 synthetic standards.

The STDI, the authors say that it was used to quantify, I do not think that this is a quantification by STDI, since, for this, several STDIs had to be used, throughout the run. Since the STDI, it is a compound that must show a behavior similar to the analytes.

I believe that the authors used the STDI to control for fluctuations in the methodology, however, the peaks identified are very far from the peak of the STDI, which is not suitable for this control. What I recommend is that the STDI used to be used to calculate series A, not for B. Therefore, I propose that if they want to present a quantification they could be adapted to STDE, presenting the equation of the calibration curve. Otherwise, the authors can only present the identification of the chromatographic peaks, not to mention quantification, in this way Figures 1 and 2 and Tables 1 and 2 are valid.

In figure 4 they present the quantitative results, which I assume are in area accounts or abundance normalized with STDI. Which is not an adequate quantification, since the calibration curves are not presented. The quantification by normalization of areas has its limitations, especially with the low resolution of the peaks.

Reviewer 3 Report

This is a good paper, well executed and well written. Congratulations.

I have just a small comments most probably to consider for the discussion section.

Is it possible to consider the population aspect of the variability of the cuticle chemistry in your discussion? I suppose that there are different populations of Bactrocera tryoni in Australia? if so, can you insert such aspect in your discussion section?

Round 2

Reviewer 1 Report

Thank you for the replay and thank you for the minor corrections made in the manuscript. 

In my opinion, considering the high reputation of the journal "Molecules" the paper is not suitable for publication